# Observation of Chiral-selective room-temperature phosphorescence enhancement via chirality-dependent energy transfer

Biao Chen [1] ✉, Wenhuan Huang[1,2] & Guoqing Zhang [1,2] ✉

Pure organic room-temperature phosphorescence (RTP), particularly from guest-host doped systems, has seen exponential growth in the last several years due to their high modulation flexibility, and yet challenges remain with respect to mechanistic elucidations and advantageous applications. Here we show that by constructing guest-host doped RTP systems from chiral components, namely, chiral amino compound-modified phthalimide hosts and naphthalimide guests, a chiral-selective RTP enhancement phenomenon can be observed. For example, *R*-enantiomeric guests in *R*-enantiomeric hosts produce strong red RTP afterglow while no appreciable RTP could be observed in the *S-R* guest-host counterpart. An unprecedented RTP intensity difference > $10^2$ folds with the ability to distinguish an enantiomeric excess of 98% could be achieved. Temperature-dependent measurements suggest that a chirality-dependent energy transfer process may be involved in the observed phenomenon, which can be harnessed to extend the RTP application to the chiral recognition of amino compounds, such as amino alcohols.

Organic luminogens have received tremendous attention due to revived interest in long-lived room-temperature phosphorescence (RTP) in the recent decade. New RTP-based applications, including next-generation optoelectronics, data encryption, anti-counterfeiting and high-contrast bioimaging, have been developed by virtue of the spin multiplicity and long lifetime of triplet excitons from organic molecules[1–3]. A subfield of organic RTP blessed with versatile molecular structures and limitless combinations, doped RTP systems have made significant progress in advancing the RTP field[4–10]. However, a comprehensive and in-depth understanding of the complex and possibly multiphase solid-state photophysics remains a major challenge. Moreover, since most of the current applications of doped RTP materials thus far are mainly focused on encryption and anti-counterfeiting, expanding new applications with unique advantages of organic doped RTP, such as molecular-chirality-related sensing owing to its slow decay kinetics and triplet-triplet energy transfer presence, is warranted for sustained interest[11–13]. Unfortunately, RTP is currently inefficient in chirality-related sensing applications; RTP chiral selectors are usually inclusion complexes (e.g., cyclodextrin[14–16]) or solid-state

substrates (e.g., menthol[17]), inducing the RTP of enantiomer analytes by restricting the nonradiative transitions and resistance to quenchers such as diffusing molecular oxygen; nevertheless, the signal distinction, albeit scarce in examples, is very subtle with only minor differences (e.g., less than a one-fold difference) showing up in lifetimes between the two enantiomeric phosphors. The subtlety is not surprising since such discrimination relies solely on the limited environmental disparity effects (e.g., Wallach' rule[18]), which mainly dictate the complementary interactions between chemical groups of the analyte and substrate.

Herein, we report that by employing a doped chiral RTP system consisting of a brominated phthalimide host and a naphthalimide guest (Fig. 1a, It has to be noted that the phrase host in this study is referred to as an excited-state energy donor, the majority species, and guest the acceptor and minor species.), we observed a strong chiral-selective RTP enhancement (CPE) effect, where enantiomeric guests in enantiomeric hosts with identical chirality produce strong red RTP while no appreciable RTP could be observed in the matrix components with opposite chiralities. It is found that the spectral difference in RTP

[1]Hefei National Research Center for Physical Sciences at the Microscale, University of Science and Technology of China, Hefei, Anhui, China. [2]Hefei National Laboratory, University of Science and Technology of China, Hefei, Anhui, China. ✉e-mail: biaochen@ustc.edu.cn; gzhang@ustc.edu.cn

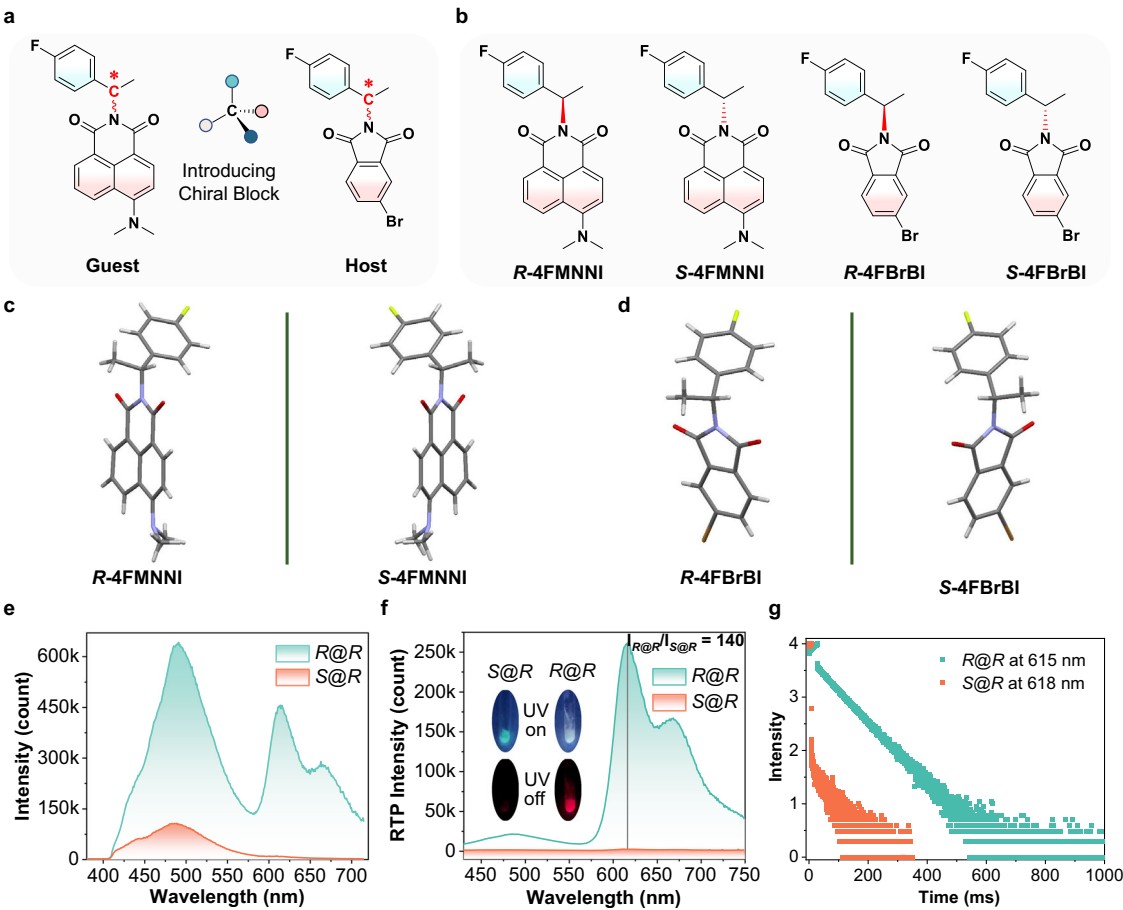

**Fig. 1 | Chiral-selective RTP enhancement (CPE) with ep > $10^2$. a** Rational design of chiral organic phosphors for doped guest–host RTP systems. The chiral amino as chiral carbon block is introduced to the naphthalimide guest and phthalimide host, the heavy-atom Br in the host promotes the ISC process and makes the host an excellent triplet energy donor. **b** Molecular structures of four chiral amino-modified compounds for constructing doped RTP systems. **c** Molecular configurations obtained from single-crystal X-ray diffraction (SC-XRD) for guests **R−4FMNNI** and **S−4FMNNI**. **d** SC-XRD structures for hosts **R−4FBrBI** and **S−4FBrBI**. **e** Steady-state photoluminescence (PL) spectra of two chiral guests ($w/w$ 10 ppm, cyan for **R−4FMNNI** and bright red for **S−4FMNNI**) in the **R−4FBrBI** host solid (namely, **R@R** and **S@R**) at 298 K ($\lambda_{ex}$ = 365 nm). **f** Delayed emission (DE, $\Delta t$ = 0.1 ms) spectra of two chiral guests ($w/w$ 10 ppm) in the **R−4FBrBI** solid at 298 K ($\lambda_{ex}$ = 424 nm). Inset: photographs of combinations of two guests in **R−4FBrBI** during and immediately after 365-nm light irradiation. ep value = $I_{R@R}/I_{S@R}$. I represent the highest intensity. **g** Time-resolved RTP emission for **R−4FMNNI** and **S−4FMNNI** in solid **R−4FBrBI** ($w/w$ 10 ppm).

signals can be augmented to an unprecedented level on the order of $10^2$ folds presumably due to the presence of chirality-dependent host-guest energy transfer in the excited state. We also demonstrated that the CPE phenomenon can be applied to chiral recognition of an amino alcohol with high enantioselectivity. As a result, the CPE supplies an angle for understanding the mechanism of guest–host RTP systems, and for expanding the potential application of RTP.

## Results

### Design and discovery

Two chiral amino compounds are chemically modified into naphthalimides **R−4FMNNI** and **S−4FMNNI** (Fig. 1b), as guest molecules and two host compounds (**R−4FBrBI** and **S−4FBrBI**), which are characterized by $^1$H- and $^{13}$C-NMR spectra, high-resolution mass spectrometry (HRMS, Supplementary Figs. 30–47), elemental analysis (EA) and chiral high-performance liquid chromatography (CHPLC), respectively. The CHPLC traces (Supplementary Figs. 3–4) demonstrate that the model compounds are chiral molecules with high purity >99.5% and high ee values >99.9%. Crystal parameters (Fig. 1c, d) obtained from single-crystal X-ray diffraction (SC-XRD) reveal that these molecules share the same monoclinic, space group ($P2_1$), and similar dihedral angles between the phenyl and imide planes (109.28°–113.32°), suggesting high structural similarity in the solid

state. The CD spectra (Supplementary Fig. 7) were used to determine their chirality with an absolute CD signal of 10–20 mdeg (0.1 mM). The absorption (Supplementary Fig. 8) and emission spectra in dilute dichloromethane ($CH_2Cl_2$) show that the **4FMNNI** set exhibits intense (Supplementary Fig. 9, Supplementary Table 1) fluorescence while **4FBrBI** has no discernible emission, likely due to the heavy-atom effect[19]. At 77 K, **4FBrBI** displays phosphorescence with a lifetime of several milliseconds, while **4FMNNI** has no appreciable delayed emission, suggesting its low intrinsic phosphorescence yield.

The sp$^3$-linked structural similarity of the host and guest makes them suitable for the configuration of guest–host doped systems[20]. According to Beard and Luther et al.[21], solution-processed molecular assemblies retain their chiral selectivity in the solid-state; we thus prepared a doped chiral RTP system with **R−4FBrBI** as a solid-state host directly from solvent evaporation without additional engineering processing. Cyan fluorescence (circa 485 nm, $\tau$ = 3.62 ns) and red RTP (circa 615 nm) arise simultaneously, constituting a visibly white emission color when **R−4FMNNI** (10 ppm, $w/w$) is coevaporated with the host medium (**R@R**, Fig. 1e). Surprisingly, when its enantiomer (**S−4FMNNI**) is applied to **R−4FBrBI**, only fluorescence occurs with no discernable RTP. The delayed emissions (DE) show that the enantiomeric RTP enhancement ratio (ep = $I_{R@R}/I_{S@R}$) is 140 (Fig. 1f), with an **R@R** RTP lifetime of 73.31 ms (Fig. 1g, Supplementary Table 2).

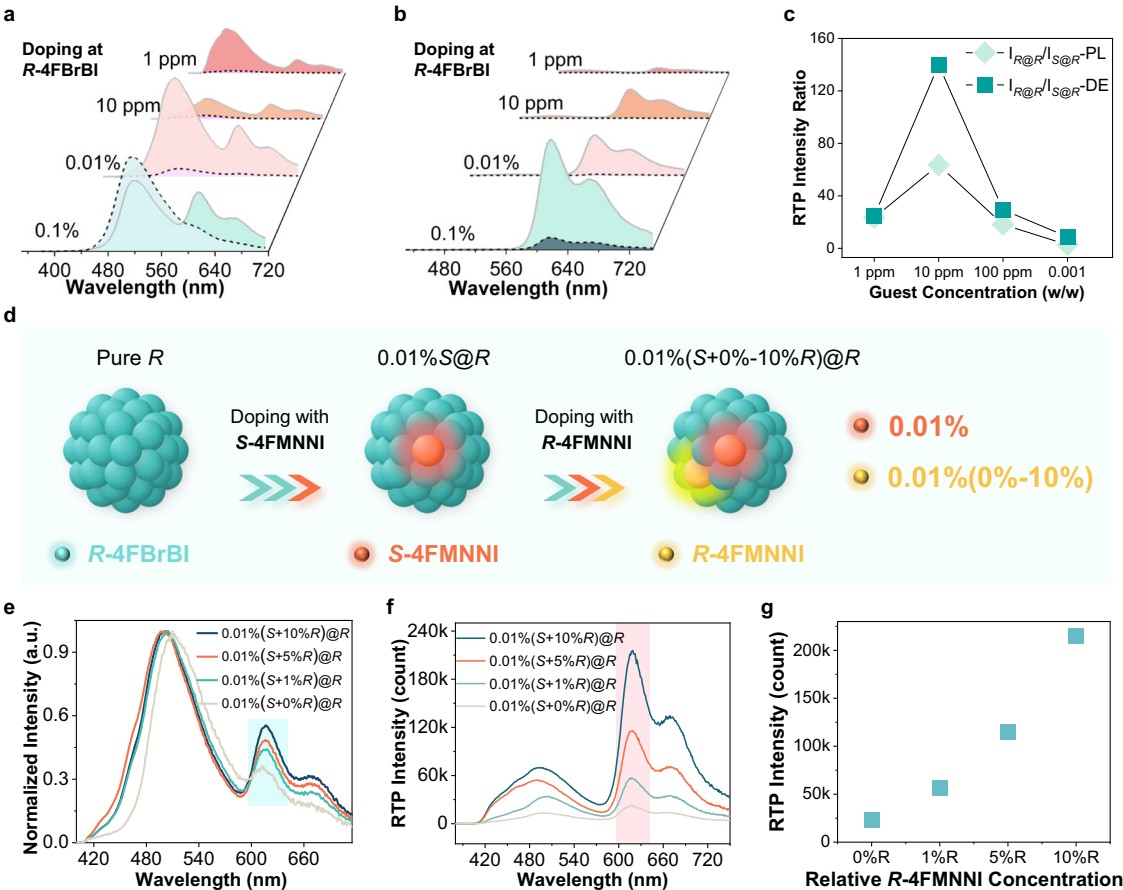

**Fig. 2 | The influences of different guest−host ratios and enantiomeric excess (ee) values. a** Steady-state PL spectra of **R**-4FMNNI (solid borderline) or **S**-4FMNNI (dash borderline) dopants ($w/w$ 1 ppm-0.1%) in **R**-4FBrBI solid (namely, **R@R** and **S@R**) in air at 298 K ($\lambda_{ex}$ = 365 nm). **b** Delayed emission (DE, $\Delta t$ = 0.1 ms) spectra of **R**−4FMNNI (dash borderline) and **S**-4FMNNI (solid borderline) guests ($w/w$ 1 ppm-0.1%) in **R**-4FBrBI solid in air at 298 K ($\lambda_{ex}$ = 424 nm). **c** The ep ($I_{R@R}/I_{S@R}$) value (**I** represent the intensity of PL emission at 620 nm or DE emission at 615 nm) vs. concentration of guest dopants ($w/w$ = 1 ppm-0.1%). **d** Measurements preparation schematic diagram for samples in **e**−**g** with different ee values: Pure **R**-4FBrBI first doped with **S**−4FMNNI (0.01%) and then doped with **R**−4FMNNI (0.01%(0%, 1%, 5%, or 10%)) to afford four samples with ee values of 100%, 98.02%, 90.48% and 81.82% (a total doping concentration of approximately 100 ppm). **e** Steady-state PL spectra of **R**−4FMNNI dopants (0–10%) in 0.01% (100 ppm) **S**−4FMNNI@**R**−4FBrBI solid in air at 298 K ($\lambda_{ex}$ = 365 nm). **f** Delayed emission (DE, $\Delta t$ = 0.1 ms) spectra of **R**−4FMNNI dopants (0–10%) in 0.01% **S**−4FMNNI@**R**−4FBrBI solid in air at 298 K ($\lambda_{ex}$ = 365 nm). **g** Intensity of DE emission at 618 nm in **f** vs. percentage of **R**−4FMNNI dopants (0–10%).

Subtraction of the background host signal from the total RTP will give a higher ep, especially when excited at 424 nm (Supplementary Fig. 11), thus yielding a dramatic chiral-selective RTP enhancement (CPE) in organic chiral solids (OSC). The DE spectra excited at 365 nm display lower ep values due to host RTP interference (Supplementary Fig. 11). The host **S**−4FBrBI (the enantiomer of **R**−4FBrBI) can also achieve enantioselectivity as shown in Supplementary Figs. 12–13, where a mirror-image relation of RTP responses is observed for the enantiomers of **4FMNNI**, i.e., ep ($I_{S@S}/I_{R@S}$) = 155 with an RTP lifetime of 70.55 ms for S@S, confirming the consistency of the CPE using the chiral guest-host system.

**Influences of guest−host ratios and enantiomeric excess values**
The guest−host ratio and its influences on CPE were then investigated. The PL spectra are presented in Fig. 2a and Supplementary Fig. 14a. When the host and the guest possess the same chirality (i.e., **R@R** or **S@S**), dramatically stronger RTP in the wavelength range of 600–720 nm emerges compared to samples of opposite chiralities (**S@R** or **R@S**), resulting in ep typically >10 (Fig. 2c and Supplementary Fig. 14d). The ep value of a guest−host doping of 0.1% is found to be <3 because of the intense guest fluorescence spectrum extending into the red region and elevating the baseline. On the other hand, **S@R** (or **R@S**) also exhibits appreciable RTP intensity when doped at high

concentrations (>100 ppm), possibly from guest aggregate RTP formation[22] or contribution from the enantiomeric impurity limited by current instrument detections. The highest ep is obtained in a doping ratio range of 1–100 ppm (e.g., ep > 55 for the 10-ppm sample), which is consistent with the DE spectra in Fig. 2b. Furthermore, the DE spectra present higher ep values (Fig. 2c, from 25 to 155 corresponding to guest−host ratios from 1 to 100 ppm) than steady-state PL by eliminating short-lived luminescence interference (e.g., guest fluorescence and host phosphorescence), showcasing the prominent advantage of RTP. We found that the guest doping concentration of 1 ppm is sufficient for a highly enantioselective RTP enhancement (CPE) of 25–32-fold, while many chiral recognition methods, such as enantioselective fluorescence enhancement[23], demand that the enantiomer analyte be at least in equal proportion or dozens of times in excess to the chiral selector[24]. With its high sensitivity, the guest−host RTP recognition method could be applied to ultratrace chiral analysis.

To test the chiral differentiation potential with CPE, by using **S**−4FMNNI-doped **R**−4FBrBI ($w/w$ 0.01%) as an example, the sample preparation process is displayed in Fig. 2d. The RTP spectroscopic between 600–720 nm range in PL spectra changes with **R**−4FMNNI at varying enantiomeric compositions were recorded (Fig. 2e and Supplementary Fig. 15), where $S + 0\%R$, $S + 1\%R$, $S + 5\%R$ and $S + 10\%R$ represent that the ee values ([S]-[R])/([S] + [R]) of the partially racemic

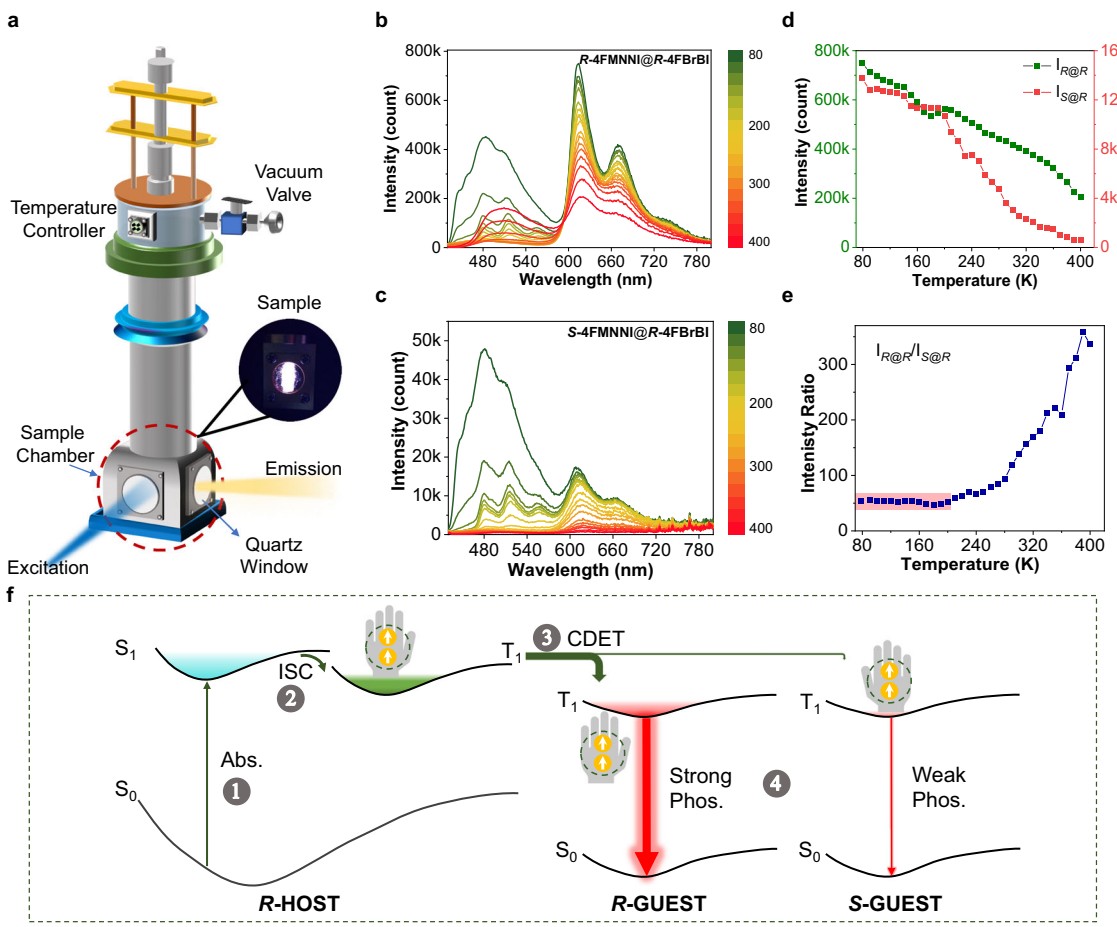

**Fig. 3 | Temperature-dependent spectroscopy exploration and mechanism perspective. a** Cryostat equipment for measuring temperature-dependent delayed emission (TDDE) spectra. **b, c** Temperature-dependent delayed emission ($\Delta t = 0.1$ ms, TDDE) spectra of **R**−4FMNNI (**b**) and **S**−4FMNNI (**c**) in the **R**−4FBrBI solid (namely, **R@R** and **S@R**, 10 ppm, $\lambda_{ex} = 424$ nm), from 80 K to 400 K with 10 K intervals (for simplicity, **b**, **c** only display 20 K intervals, 10 K interval spectra are shown in Figures S15-S16). **d** Intensity at 615 nm obtained from **b** and **c**. vs. temperature. **e** The ep ($I_{R@R}/I_{S@R}$) values vs. temperature. Note that ep remains almost constant below 200 K with a high value of approximately 50. **f** A plausible chirality-dependent energy transfer (CDET) process for the chiral-selective RTP enhancement (CPE) phenomenon. When the samples are excited, the photon energy is first converted to singlet excited states of host molecules since they are the overwhelming majority (Step 1). The host singlet excitons rapidly transform

to triplet ones (indicated by parallel electron spins shown in circle) through the efficient ISC process, which is enhanced by strong spin-orbit effect of the internal heavy-atom Br (Step 2). Subsequently, energy transfer occurs from the host triplet to the guest triplet, and the efficiency is dependent on chirality, namely, the chirality-dependent energy transfer process, which is much more effective when the enantiomeric guest and host have identical chirality (e.g., **R-HOST** and **R-GUEST**) compared to opposite chirality (**R-HOST** and **S-GUEST**), and produces more triplet excitons in **R-GUEST** (Step 3). The guest molecule traps the triplet excitons and emit strong phosphorescence from $T_1$ to $S_0$ via radiative transition, and vice versa, no appreciable phosphorescence could be observed in **S-GUEST** (Step 4). Abs: absorption, ISC: intersystem crossing, CDET: chirality-dependent energy transfer. Higher excited states $S_n$ and $T_n$ and internal conversion (IC) are omitted for brevity.

guest analyte **4FMNNI** compounds are 100%, 98.02%, 90.48% and 81.82%, respectively. As can be observed in the DE spectra (Fig. 2f), the RTP intensity is enhanced with a higher **R**-guest proportion. Remarkably, there is a measurable RTP improvement (2–3 × by tuning the excitation wavelength, Fig. 2g and Supplementary Fig. 16) between enantiomeric excess (ee) values of 100 and 98%, implying the successful distinguishability for an enantiomer (**R−4FMNNI** in this case) lower than 2% from the racemate, i.e., the system can be used to resolve the enantiomeric composition and an enantiomer impurity with an ee of 98%.

## Mechanistic investigations

Conventionally, the mechanism of chiral recognition with photoluminescence assumes that molecules with matching chirality exhibit stronger nonbonding interactions, which rigidifies the analyte-substrate pair and diminishes the nonradiative decay rate $k_{nr}$. As a result, one would expect to see ep approaching ~1 when the measurements were performed at cryogenic temperatures. Therefore, we

collected temperature-dependent delayed emission (TDDE) spectra of **R@R** and **S@R**, respectively (Fig. 3b, c, Supplementary Figs. 17–18), using the cryostat equipment shown in Fig. 3a. The phosphorescence intensity and intensity-ratio ep changes as a function of temperature at the emission maximum wavelength ($\lambda_{RTP} = 615$ nm) for both systems are shown in Fig. 3d, e. Surprisingly, ep remains almost constant below 200 K with a high value of ~50 even near 77 K (Fig. 3e), suggesting that the $k_{nr}$ factor is not the primary reason for CPE. Beyond 200 K, however, the continuous upward trend of the ep value vs. temperature suggests that the **S@R** system becomes more susceptible to quenching at high temperatures. In comparison, using nonenergy transfer chiral media, such as **R**-camphor (free chiral amine is a quenching medium and cannot be used), produces almost identical phosphorescence profiles (ep ~ 0.56) at 77 K (Supplementary Fig. 19), which is also consistent with the literature[14–17]. To gain more insights, we also compared the powder XRD patterns, wide-temperature range differential scanning calorimetry (DSC), and TGA (Supplementary Figs. 20–22), and we found almost no difference between these two

groups, indicating that no phase transitions or other intermolecular interactions participated in the CPE. Given that RTP from guest–host systems relies on excited-state energy transfer[25,26], it is quite possible that exciton migration from the host molecule to the guest could be chiral selective. Thus, we speculate that a chirality-dependent energy transfer process (CDET) may be responsible for the CPE issue, which along with the $k_{nr}$ factor, causes extraordinarily high ep at RT and higher temperatures.

Compiling these results together, we proposed a plausible mechanism for the chiral-selective RTP enhancement (CPE) issue (Fig. 3f). Four stages are involved: (1) absorption: the photon energy is mostly absorbed by host molecules as they occupy overall majority in the doped samples (>99.999% for 10 ppm, for example) and produce high energy excited singlet states ($S_n$ is omitted for simplicity), following by internal conversion (IC) to lowest singlet state ($S_1$) of hosts; (2) ISC: lowest singlet state ($S_1$) of host effectively transforms to triplet states through ISC process, which is enhanced by strong spin-orbit effect of heavy-atom Br, note that the host triplet is an optical dark state, which produces no luminescence at room temperature (as seen before in Supplementary Figs. 9 and 11); (3) chirality-dependent energy transfer (CDET) process: energy transfer from host triplet states (the presence of which is indicated in the singlet-oxygen sensitization experiment shown in Supplementary Fig. 28) to guest triplet states ($T_1$ or $T_n$) is extraordinarily ordinary in doped RTP systems, for these chiral doped system, the efficiency is dependent on chirality and much more effective when enantiomeric guest and host possess same chirality, such as *R*-HOST and *R*-GUEST, possible for similar transition dipoles or shorter intermolecular space, as a result, much more triplet excitons accumulated in *R*-GUEST compared to *S*-GUEST; (4) RTP emission: the triplet excitons trapped in guest molecules eventually dissipate energy from the lowest triplet state ($T_1$), when decay to ground state via radiative transition, register as room-temperature phosphorescence in air, guest–host system with identical chirality produces much stronger RTP (100×) for more guest triplet excitons, which manifests as the chiral-selective RTP enhancement (CPE) phenomenon we observed.

To experimentally verify the proposed CDET mechanism, we measured the phosphorescence lifetimes of various samples at both 77 K and at room temperature (Supplementary Fig. 27), where the decay kinetics unambiguously reveal how host triplet excitons can be more selectively depleted by guest molecules of the same chirality but are less unaffected by guest molecules of the opposite chirality. As can be seen from Supplementary Fig. 27a, the pure host of *R*−4FBrBI exhibits single-exponential decay kinetics monitoring at 420 nm (phosphorescence emission belonging to *R*−4FBrBI) with an apparent lifetime of 5.4 ms at 77 K. However, the phosphorescence lifetime is substantially reduced when a trace amount (10 ppm) of guest *R* −4FMNNI is present, yielding bi-exponential decay kinetics (Supplementary Table 7, $\tau_1 = 0.4$ ms and $\tau_2 = 3.5$ ms) with a pre-exponential-weighted average lifetime of 3.0 ms. To show that the energy-transfer processes is indeed chirality-dependent, we also measured the decay kinetics of *S*−4FMNNI@*R*−4FBrBI (10 ppm) monitoring at 420 nm. Surprisingly, the decay kinetics are more similar to those found for the pure host *R*−4FBrBI (Supplementary Fig. 27a), suggesting less effective triplet-triplet energy-transfer processes. In addition, these results also give direct evidence for the Dexter-type CDET, since a dominant long-range Förster ET should not distinguish chirality, a short-range effect in this case. Furthermore, it is well known that triplet-triplet energy transfer cannot proceed with the Förster type, given the vanishingly small transition dipole moments[27]. In the current experimental condition (crystalline state), the donor and acceptor molecules are clearly separated by less than 1 nm, also creating the necessary condition for the Dexter process. We also performed the same measurements at room temperature. Although triplet-state quenching was also noted by guest molecules, the apparent decay kinetics for these samples are non-exponential (Supplementary Fig. 27b), which is not

unusual for organic solids at elevated temperatures. We attribute the observed complexity at room temperature to enhanced molecular motions and various back-population (i.e., reverse intersystem crossing) pathways.

## Application

Finally, the guest–host RTP chiral recognition method based on CPE was extended to a different amino compound. Chiral amino alcohols have been extensively used in medicine, racemate resolution and asymmetric synthesis[28,29]. Enantioselective recognition of amino alcohols has thus attracted significant research attention in recent years[30]. Here, we show that the CPE effect could be applied to chiral amino alcohol recognition using 2-phenylglycinol (Pg) as an example. Following the same protocol, four chiral derivatives (Fig. 4a, b, *R*-PgMNNI and *S*-PgMNNI as guest molecules) and two racemates (Supplementary Figs. 5–6, Supplementary Figs. 48–65) were synthesized and characterized. When doped at a concentration of 100 ppm, PL spectra (Fig. 4c) reveal that the host-guest system with the same chirality (*R*@*R* and *S*@*S*) exhibits more pronounced RTP enhancement over *S*@*R* or *R*@*S*, where even the naked eye can readily discriminate the enantioselective luminescence under UV excitation, since *S*@*R* (or *R*@*S*) shows cyan emission while the other two are either orange or pink (photos in Fig. 4e). The DE spectra (Fig. 4d) also registered an RTP enantioselective enhancement >10-fold ($I_{R@R}/I_{S@R} = 11.5$ and $I_{S@S}/I_{R@S} = 12.7$) with a longer lifetime (Fig. 4f, 63.25 ms for *S*@*S*, 63.36 ms for *R*@*R*, Supplementary Table 3).

## Discussion

In summary, we designed and constructed guest–host organic chiral solid (OCS) systems with two sets (4F and Pg series) of chiral imide molecules, and observed unprecedented enantioselective RTP enhancement (CPE) values (>10² at room temperature) from the OCSs. A chirality-dependent energy-transfer (CDET) mechanism was proposed for the CPE issue based on experimental results. We also showed that RTP could be applied to enantioselective recognition, including chiral amino alcohols with high sensitivity and ultratrace analysis. The observation reported here may pave the way for a deeper understanding of molecular photophysics and exciton migration in chiral organic solids, and supply methods for phosphorescence manipulation and chiral detection.outstand

## Methods
### Materials
4-Bromophthalic anhydride, 1-(4-fluorophenyl) ethanamine and dimethylamine were purchased from Shanghai Aladdin Bio-Chem Technology Co., Ltd. All other reagents and solvents were obtained from Energy Chemicals and were used as received. Water was deionized with a Milli-Q SP reagent water system (Millipore) to a specific resistivity of 18.2 MΩ.cm. Column chromatography was performed using silica gel 60 (230–400 mesh) with the indicated solvents.

### Instrumentation
NMR spectra were recorded on a Bruker AV400 NMR spectrometer operated in Fourier transform mode, and NMR spectra were recorded at 400 MHz for ¹H and 101 MHz for ¹³C. Electrospray ionization (ESI) mass spectra were recorded on an Acquity UPLC-Xevo G2 QT mass spectrometer (Waters). Elemental analysis (EA) was performed on an Elementar Vario MICRO elemental analyzer. Gel filtration chromatography was performed using a chiral column (CHIRALPAK® AD-H 0.46 cm I.D. x25 cm × 5 μm, DAICEL) conjugated to an Agilent 1260 Infinite HPLC system. Prior to characterization, each sample was purified via a 0.45 μm filter to remove any aggregates. The flow rate was fixed at a rate of 0.5 mL/min, the injection volume was 10 μL and each sample was run for 50 min. The absorption wavelength used was set at 300 nm and 370 nm. Ethanol and n-hexane were used as the eluting

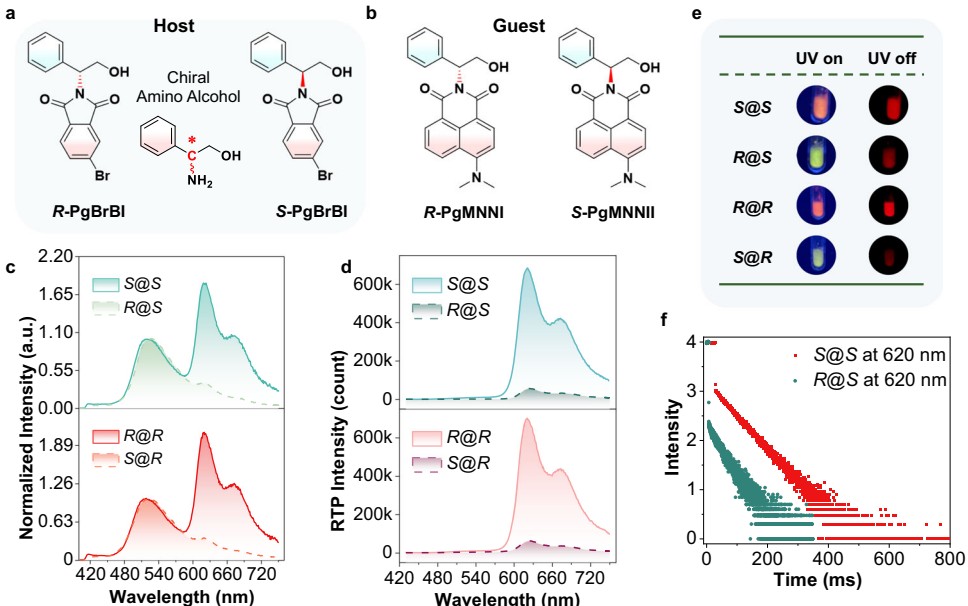

**Fig. 4 | Chiral amino alcohol recognition with CPE. a, b** Molecular structures of chiral amino alcohol derivatives: chemically modified chiral analyte guests (**b**, **R-PgMNNI** and **S-PgMNNI**) and chiral selector hosts (**a**, **R-PgBrBI** and **S-PgBrBI**) molecules. **c** Normalized steady-state PL spectra of two chiral guests ($w/w = 100$ ppm) in two chiral host solids (for example, **R@R** represents **R-PgMNNI@R-PgBrBI**, guest molecule in front of @) at 298 K ($\lambda_{ex} = 380$ nm). **d** Delayed emission (DE, $\Delta t = 0.1$ ms) spectra of two guests ($w/w$ 100 ppm) in two host solids at 298 K ($\lambda_{ex} = 424$ nm). **e** Photographs of combinations of two guests in **R-PgBrBI** or **S-PgBrBI** during and immediately after 365-nm light irradiation. **f** Time-resolved emission intensity at 620 nm of **R-PgMNNI** and **S-PgMNNI** in solid-state **R-PgBrBI**.

solvents. UV/Vis absorption spectra were recorded on a PerkinElmer Lambda 465 UV–Vis spectrometer. Steady-state emission spectra were recorded on a Horiba FluoroMax-4 spectrofluorometer (Horiba Scientific). The light source (xenon lamp) that supplies UV excitation is focused onto the entrance slit of the excitation monochromator with an elliptical mirror. According to the manufacturer, the light source is a vertically mounted 150-W ozone-free cw xenon arc lamp. The distance between excitation and the sample was -120 cm, and the integration time is 0.1 s. Fluorescence lifetime data were acquired with a 1 MHz LED laser with an excitation peak at 372 nm (NanoLED-370). Phosphorescence lifetime data were acquired with a LED laser with an excitation peak at 374 nm (SpectraLED-370) or at 344 nm (SpectraLED-340). Lifetime data were analyzed with Data Station v6.6 (Horiba Scientific). Photographs were taken by an iPhone 13 camera. The delayed emission spectra were recorded on a Horiba FluoroMax-4 spectrofluorometer as well, using a 10-W xenon flash lamp as the excitation source with an integration time of 0.1 s. The absolute photoluminescence quantum yields were measured on a Horiba FluoroMax-4 spectrofluorometer (Horiba Scientific) with an integrating-sphere (Labsphere Inc.). Single-crystal samples were obtained by slow evaporative crystallization of compounds in dichloromethane, chloroform or n-hexane, and single-crystal data were collected on a Bruker Smart APEXII CCD diffractometer using graphite monochromated Cu-Kα radiation ($\lambda = 1.54178$ Å). Podwer XRD was measured on a Multifunctional Rotating-anode X-ray Diffractometer. TGA was measured on a NETZSCH TG209F1 Libra. from 30–600 °C (10 K/min) in $N_2$. DSC was measured on a DSC Q2000 V24.10 Build 122 from −90 to 180 °C at 10.0 °C/min.

### Sample preparation
All guest–host samples were prepared by serial dilution and subsequent evaporative drying from their dichloromethane ($CH_2Cl_2$) solutions with trace hexane if needed (different solvents may affect photoluminescence). Specifically, a fixed amount of the host compound (100.00 mg) was placed in a 5-mL vial, to which a dichloromethane solution of the guest with various concentrations and volumes was added. The liquid was vigorously mixed to yield a clear

solution, which was then allowed to evaporate under ambient conditions for overnight. The crystalline solid was then collected and thoroughly dried in vacuum for at least 48 h prior to optical measurements. An example of the serial dilution of the guest molecule is provided: a stock solution (100 μL with a concentration of 10.00 mg/mL) in dichloromethane was withdrawn and pipetted into a volumetric flask (10 mL), which yielded a dichloromethane solution with a concentration of 0.1 mg/mL. For each dilution, a concentration reduction by a factor of 100 can be achieved. The exact mass of the guest molecule can be calculated by withdrawing from one of the four stock solutions (10.00, 0.1, 0.001, and 0.00001 mg/mL) in preparing the guest–host mixed solid samples. The pure host solid sample (control) was obtained by the same procedure by adding the same amount of dichloromethane without guest molecules to exclude any possibility of contaminant influence from solvents. In particular, for the three-component doped sample such as ones used in Fig. 2d–g, a fixed quantitify of the pure host **R−4FBrBI** (100.00 mg) was placed in a 5-mL vial, to which the dichloromethane solution of **S−4FMNNI** (0.1 mL, 0.1 mg/mL) was added, followed by the additon of a dichloromethane solution of **R−4FMNNI** of various concentrations, and was finished by adding dichloromethane to adjust the solvent amount of all samples to be exactly the same. It must be noted that all the mixed solid samples are calculated by mass fraction and that the molecular weight of the guest (**R−4FMNNI**, **S−4FMNNI**, **R-PgMNNI** and **S-PgMNNI**) is greater than that of the host **BrBI**, so that the molar fraction will be numerically smaller than the mass fraction, i.e., 10 ppm ($10 \times 10^{-9}$) by mass ratio of **MNNI** is the equivalent of 9.58 ppm by molar ratio.

### Data availability
All relevant data generated in this study are provided in the supplementary information and also are available from the authors upon request. Source data are available. Source data are provided with this paper. CCDC 2150413-2150416 contains the supplementary crystallographic data for this paper. These data are provided free of charge by The Cambridge Crystallographic Data Centre. Source data are provided with this paper.

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

## Acknowledgements

We thank the National Natural Science Foundation (22003063 and 22273097 to B.C. and 21975238 to G.Z.) for financial support. We also thank Prof. Yi Luo for his helpful discussion. We are grateful to Ms. Qingqing Yang for her kind help and her artistic retouching of the figures.

## Author contributions

B.C. conceived the project and designed the experiments. B.C. synthesized all compounds. B.C. and W.H. contributed to optical characterizations. B.C. optimized HPLC and grew crystals. B.C. solved crystal structures. B.C. and G.Z. wrote the manuscript. B.C., W.H., and G.Z. discussed the results and edited the manuscript. B.C. and G.Z. supervised the project.

## Competing interests

The authors declare no competing interests.
