## [Peer Review File · Nature Communications]

Observation of Chiral-Selective Room-Temperature
Phosphorescence Enhancement via Chirality-Dependent
Energy TransferReviewers' Comments:

Reviewer #1:

Remarks to the Author:

Chen et al. proposed a concept of chirality-dependent energy transfer to explain the unusual behavior of what they claimed the chiral-selective RTP enhancement. Practically, the authors provided ample steady-state/time-resolved measurements on the topic molecules in attempts to rationalize their emissive behavior/properties, and finally came up with the mechanism. After careful examination, unfortunately, I have to conclude that the proposed mechanism was fundamentally groundless and unconvincing, where some corresponding assignments are very far-fetched. Therefore, I do not recommend the publication of this manuscript at current format. Further experiments and convincing assignments and hence mechanisms proposed are required. Comments are listed below.

1. The major flaw of the experiment and hence mechanism lies in the fact that the host molecules cannot be excited with either 365nm or 424nm. As shown in In the Figure S6, the absorption onset of the host molecules is near 350 nm. Thus, the mechanism has nothing to do with the host after being excited at these wavelengths applied in this study. The excitation wavelength should be appropriate before the authors can provide any reasonable results and rational discussion.

2. The authors claimed that the red emission in Figure 1f originated from phosphorescence of the guest molecules, and one can see that the emission shows apparent vibronic structure. However, in comparison to the delay emission spectra shown in Figure S7(d), the emission pattern is drastically different.

3. Continuing comment #2, it is also critical to determine where the blue emission come from. Although the blue emission exhibits nanosecond population decay, which may support its assignment as fluorescence, upon exciting the host-guest mixtures at 365 nm, the guest molecules would be directly excited and display fluorescence as well. However, the emission onsets of them (Figure S7) are different from the blue band region in Figure 1f. Therefore, more likely, the blue band emission comes from the host since their onsets are the same (see Figure S7(c)). Also, according to my rough calculation, the energy transfer efficiency of the host is estimated to be ~100%. Accordingly, there should be nearly no population for the host in the excited state (triplet state) to observe the corresponding phosphorescence. Excitation spectra monitoring at the dual emission bands are absolutely necessary and the authors should check if the excitation and absorption spectra resemble in spectral profile and onset.

4. In Figure S8(c) and S8(d), owing to the large Stokes shift and heavy-atom effect, the host molecules should exhibit phosphorescence in 77K. However, it is very weird that the prompt and delay ($t=0.1\text{ms}$) emission spectra display different emission onset.

5. The above concerns that I raised are all serious enough to affect the proposed mechanism. And the I strongly encourage the authors to carry out not only the emission spectra but also their corresponding excitation spectra before making any spectral assignment, so that the proposed mechanism can be clarified, to a large extent.

Reviewer #2:

Remarks to the Author:

In this manuscript, the authors report observation of unprecedented enantioselective RTP enhancement (CPE) values with 100 at room temperature from the organic chiral solid designed by them. This is the very good results I agree, which should be suitable for Nature Commun. The authors designed two chiral amino compounds naphthalimides R-4FMNNI and S-4FMNNI as guest molecules and two host compounds (R-4FBrBI and S-4FBrBI). For definition of Host and Guest molecules here might confused the audience, So the authors are suggested to give the clear distribution.

The Mechanistic Investigations is much interesting. The assembly of host-guest complex might be important to explain their observation of unprecedented enantioselective RTP enhancement. Therefore, this referee suggests the authors might discuss this assembly structure of host-guest

complexes.

Its application is very interesting too. the authors give the elementary study to recognize Chiral amino alcohols. this referee realized that an RTP enantioselective enhancement of derivatives of Chiral amino alcohols is about 10-folds with longer lifetime, it is a good result of course, but less exciting than the above discussed molecules. This referee is wondering if this RTP enantioselective enhancement of host-guest molecules system would be common rules? How about amino acid derivatives?

In the end, In Fig 1, R@R, S @R et al are very confusing to the audience, which might be revised to the clear description R-molecular name or S=Molecular name as example.

Reply to Reviewers' Comments for Manuscript NCOMMS-22-41250 for Nature Communications

Reply to the report of the Reviewer 1

General comment: *Chen et al. proposed a concept of chirality-dependent energy transfer to explain the unusual behavior of what they claimed the chiral-selective RTP enhancement. Practically, the authors provided ample steady-state/time-resolved measurements on the topic molecules in attempts to rationalize their emissive behavior/properties, and finally came up with the mechanism. After careful examination, unfortunately, I have to conclude that the proposed mechanism was fundamentally groundless and unconvincing, where some corresponding assignments are very far-fetched. Therefore, I do not recommend the publication of this manuscript at current format. Further experiments and convincing assignments and hence mechanisms proposed are required. Comments are listed below.*

Author Reply: We thank the reviewer for the careful examination, the doubts raised and suggestions, which provide us an opportunity to further elaborate the mechanism and make the article more robust. We have now conducted several further experiments and more clear assignments to support the mechanism. Our point-by-point response is given below.

Comment 1: *The major flaw of the experiment and hence mechanism lies in the fact that the host molecules cannot be excited with either 365nm or 424nm. As shown in In the Figure S6, the absorption onset of the host molecules is near 350 nm. Thus, the mechanism has nothing to do with the host after being excited at these wavelengths applied in this study. The excitation wavelength should be appropriate before the authors can provide any reasonable results and rational discussion.*

Author Reply 1: We thank the reviewer for raising this important question, which is indeed the very essence and uniqueness of the system. We are happy to take this opportunity to elucidate. To fully address the concerns, we have compiled all related evidence and made lengthy discussions below. Simply speaking, the core arguments are that **the host CAN BE EXCITED with 365-nm and 424-nm light and ENERGY TRANSFER DOES HAPPEN from the host to the guest under this condition.** We thus first list these two

aspects, followed by other findings for the convenience of the editor and the reviewer.

I. Evidence from Emission Spectra: The reviewer was concerned with the fact that both 365-nm (which is not that surprising for solid state aggregates) and, oddly enough, 424-nm light (we will discuss the 424-nm problem later) can be used to excite the pure host molecular solids. We now believe that a combination of two factors contribute to the fact: absorption broadening due to molecular aggregation and the Gaussian distribution nature of the excitation beam used for RTP measurements. When the host molecules dissolve in solutions (in dichloromethane for instance) as discrete monomers, the absorption onset of the host molecules is near 350 nm as shown in Figure S6 a and c in the submitted supporting information (SI). However, the excitation band energy of molecules tends to split into a broader range when in ground-state aggregates (e.g, H and J aggregates) due to strong excitonic interactions in solid-state films (e.g., Kasha's exciton model: *Kasha M, Rawls H R, El-Bayoumi M A. The exciton model in molecular spectroscopy. Pure and applied Chemistry, 1965, 11: 371-392.*). As shown in the solid-state UV absorption spectra for hosts **4FBrBI** and **PgBrBI** (**Figure R1**): where the black dash line is set at 365 nm and the blue dash line is set at 375 nm, it is evident the onset of host solid lies below 375 nm and can be excited by the 365-nm light. (In fact, we can even see the onset starting from ~425 nm for the PgBrBI solids.)

Figure R1. Host molecules solid absorption.

In Figure R2, we show the profiles of the excitation beam under different conditions. For steady-state spectra, the light source is a 150-W ozone-free cw xenon arc lamp, with the entrance slit set at 0.5-3 nm; for delayed spectra (both delayed emission and delayed excitation spectra) the light source is a flash xenon lamp which is much lower in power, i.e., < 20-W, and the entrance slit is normally set at 5-10 nm. The linewidth is 361-367 nm for the 150-W xenon lamp used for steady-state spectra with a min. entrance slit of 0.5 nm (Figure R2a), and much more broadening is noted (350-385 nm, with an entrance slit of 10 nm, $\Delta t = 0.1$ ms) for the flash xenon lamp used for RTP spectra (Figure R2b). In fact, the broadening of the light source usually causes redshift of excitation spectra. Consequently, the combined effect of excitation band energy redshift in the solid-state aggregates and the broadening of the light source with wider excitation beam entrance could well make the samples excitable by the 365-nm light, which is used to produce the photo images in the manuscript.

Figure R2. Light source scattering using blank quartz tube in steady-state (a, entrance slit 0.5 nm) and delayed (b, entrance slit 10 nm, $\Delta t = 0.1$ ms) mode.

II. Evidence from Excitation Spectra: One of the “Golden Criteria” in luminescence spectroscopy for **evidence of energy transfer** is whether the excitation spectrum monitored at the energy acceptor (guest) emission wavelength shows excitation characteristics of the energy donor (host). Therefore, we now provide **TWO** pieces of critical evidence to eliminate the editor’s and reviewer’s concerns: **1)** we have compared the excitation spectra of the pure host solid vs. the guest-host mixture

solid (**Figure R3**), where we find that the excitation spectra of the two are almost identical in the host excitation region (same for Pg series in **Figure R4**); **2**) if a new host matrix with a different excitation profile is to be used for the same guest, the excitation spectrum would naturally shift to match that of the new host, and this is precisely what we have observed in **Figure R5**: it can be clearly seen that the RTP spectrum remains unchanged in the new host while the excitation is blue-shifted to reflect the profile of the new host. This indicates that the host is not “useless” as in the reviewer’s comment that “the mechanism has nothing to do with the host...”. If this were true and the host were “inert” to energy transfer, then we would expect to see identical excitation spectra in both hosts.

Figure R3. Solid-state excitation spectra of the guest-host (S-4FMNNI@S-4FBrBI, 10 ppm), pure host (4FBrBI), and guest (S-4FMNNI) in PMMA. The guest-host doped solid and pure host solid excitation vibronic signatures share substantial overlaps in the highlighted region while the excitation of the guest in the energy-transfer “inert” matrix PMMA shows a very different pattern (The fact that there is a broad peak is likely due to oversaturation of the PMT from high photon counts in the lower energy region, indicating that the guest excitation favors low energy photons > 370 nm).

Figure R4. Solid-state excitation spectra of the guest-host (S-PgMNNI@S-PgBrBI, 100 ppm), pure host (PgBrBI), and guest (S-PgMNNI) in PMMA.

Figure R5. (a) Structures of host S-4FBrBI and new host S-4FBI. (b) delayed ($\Delta t = 0.1$ ms) emission (black line) and excitation (red line) of S-4FMNNI@S-

4FBI (0.01% doped, quartz tube, entrance slit 10 nm) at 298 K. Host solid absorption comparison (c) and delayed excitation (DEx) spectra comparison (d) of S-4FMNNI doped at S-4FBrBI vs. at new host S-4FBI.

In addition, we also collected RTP spectra of a representative guest-host doped solid excited at different wavelengths (**Figure R6**), and found that high-energy photons (favored by host absorption) result in higher guest RTP/fluorescence ratios. If the host were again “inert” to energy transfer, we would expect that excitation at longer wavelengths (favored by guest absorption) resulted in higher guest RTP ratios. Incidentally, with this method, we were able to realize “color-tuning” with excitation wavelengths (**Figure R7**).

Figure R6. Steady-state PL spectra of 0.1% S-4FMNNI@S-4FBrBI in the solid state at 298 K with different excitation wavelengths ($\lambda_{\text{ex}} = 335\text{-}424$ nm) and intensity ratio between 615/500 nm in at different excitation wavelengths.

Figure R7. CIE Figure of 0.1% S-4FMNNI@S-4FBrBI solid at 298 K with different excitation ($\lambda_{ex} = 335-415$ nm).

We now discuss the 424-nm case (the 424 nm excitation we used in delayed emission), since solid-state excitation spectra of the guest-host are very alike to pure host except has additional fraction excitation in 400-450 nm.

The main reason is the quartz tubes we used exhibit upconversion from visible light (424 nm for example) to UV light (335 to 410 nm), such an upconverted UV band disappeared when we used different transparent tubes, for example, glass tubes only scatter the excitation beams with no upconversion (**Figure R8a**). We suspect that the quartz tubes contain certain inorganic (e.g., metal oxides) impurities responsible for the upconversion. These upconversion UV lights can excite the samples. **Figure R8a** show the broadening linewidth of the light source, scattering with a blank quartz tube or glass tube when excited at 424 nm (entrance slit 10 nm, $\Delta t = 0.1$ ms), quartz tube has additional light in 335-410 nm than glass tube. 365 nm (**b**) or 298 nm (**c**) excitation has not much upconversion light. And the proportion of stray UV light generated by tubes is also depend on entrance slit and

light source power, so for the steady-state spectra (entrance slit 0.4 nm, light power 150-W) the upconversion light is very little (**d**).

Figure R8. Light source scattering using blank quartz/glass tube in delayed (a-c, entrance slit 10 nm, $\Delta t = 0.1$ ms) and steady-state (d, entrance slit 0.4 nm) mode.

In **Figure R9a**, when we replace quartz tube to glass tube, the 424 nm region of doped sample (D-DEx) declines (it also decreased rapidly before 290 nm for the poor light transmittance of the glass tube), and the smaller the entrance slit, the smaller the proportion of 424 nm excitation (red arrow). And with small entrance slit and glass tube, the D-DEx approximates to the solid absorption spectrum of host (**Figure R9b**).

Figure R9. a, Delayed excitation (DEx) spectra of doped samples (10 ppm S-4FMNNI@S-4FBrBI at 617 nm) at different tubes or entrance slit (D and H represent doped samples and host). b, The comparison of host solid UV absorption with DEx of doped samples.

Furthermore, in **Figure R5**, the solid absorption of new host S-4FBI is blueshift compared to S-4FBrBI and the absorption onset is also blueshift to near 350 nm (**Figure R5c**), and can't be excited by 350-400 nm, which eliminate most influence of upconversion light from quartz tube, so the solid excitation spectrum of doped at S-4FBI has almost no excitation above 360 nm (**Figure R5d**), despite quartz tube is used.

There are other possible reasons for 424-nm excitation: It is very likely that there are weaker absorbing $n-\pi^*$ and/or S_0-T_1 transitions in the low-energy region compared to the much stronger $\pi-\pi^*$ states at the higher energy region, which is not uncommon for compounds (host) with both carbonyl and bromine groups. We recently published a study in *Adv. Opt. Mater.* (<https://onlinelibrary.wiley.com/doi/full/10.1002/adom.202200099>), and we found that bromine-substituted aromatic imide forms weak transition state in the visible light region, which could even be excited with green or red light due to molecular aggregation. Imide aggregates formation inhibits the 0-0 band transition and enhances the 0-1 transition in spectroscopic signature This is what we have observed for the imides used in the current study in CH_2Cl_2 at 77 K, which will be discussed in Reply 2.

Figure 2 from *Adv. Opt. Mater.* (“Broad-Band Visible-Light Excitable Room-Temperature Phosphorescence Via Polymer Site-Isolated Dye Aggregates”): Where Figure 2a shows how aromatic imide dye aggregates (which absorbs only in the UV region as discrete molecules) can be excited with much-lower energy photons. Figure 2b shows how vibronic signature is altered characteristic of aggregates formation (a progressively strongly 0-1 band vs. the 0-0 band).

Additionally, with the increase of the guest doped concentration, the guest aggregates existing at high concentration (0.1% doped especially) may participate in the generation of RTP through more complex photophysical processes, which is beyond the scope of this discussion. 424 nm excitation of doped samples exists on this condition, but is relatively much lower than the excitation of host solid at low doping concentration (**Figure R10**).

We should reiterate that our final choice of quartz tube is for its better optical transmittance than glass tube (**Figure 8d**), especially in the ultraviolet range has more complete excitation spectrum.

I hope that these results are already convincing to address the reviewer’s and editor’s serious concerns. There are, however, other new results we obtained in the past month that we would like to present to make the arguments more complete or complementary.

Figure R10. Delayed excitation (DEx) spectra of different concentration (w/w = 1 ppm-0.1%) doped samples (S-4FMNNI@S-4FBrBI at 617 nm, glass tube, entrance slit 7 nm) at 298 K.

Other evidences: We next show that the imide host molecules are the only ones inducing strong RTP, and the guest molecules **MNNI** are essentially fluorescent, not easy to produce phosphorescence or RTP at common condition. This indicates the importance of the host to guest triplet-triplet sensitization, which is key to achieve efficient phosphorescence for the guest: we repeated the guest luminescence data in DCM and conducted another set of luminescence measurement in 2-methyltetrahydrofuran (mTHF), a common solvent for studying molecular photophysics, which has the advantage of vitrification at low temperature. Emission spectra and excitation spectra are shown in **Figures R11** and **R12**, consistent with the data in DCM, except that **MNNI** has no observable phosphorescence emission.

Figure R11. a, b, Steady-state (Ss) photoluminescence (PL) spectra in 2-methyltetrahydrofuran (mTHF) (20 μ M) at 298 K (excited by absorption maxima). PL spectra (c, d) and delayed ($\Delta t = 0.1$ ms) emission (e, f) in mTHF at 77 K.

Figure R12. a, UV absorption spectra in mTHF (0.25 mM for BrBI, 0.05 mM for MNNI) at 298 K. Excitation spectra (Ex) at 298 K (b) and 77 K (c, d). Delayed ($\Delta t = 0.1$ ms) excitation spectra (DEx) at 77 K (e-f).

Furthermore, not only in solutions does **MMNI** possess barely no phosphorescence, **Figure R13** shows that **MNNI** solids exhibit strong fluorescence but also no observable RTP (**b**), with lifetimes of circa 10 ns (**c**). Note that the excitation intensity decreases below 370 nm (**d**).

Figure R13. a, b, Steady-state (Ss) PL spectra and delayed ($\Delta t = 0.1$ ms) emission of MNNI in solid state at 298 K. c, Time-resolved emission decay curves of pure MNNI solid at 298 K (excited with nanoLED-455). d, Excitation spectra (Ex) at 298 K.

Figure R14 shows that **MNNI** molecules in PMMA films (0.1% or 0.01% w/w) have strong fluorescence at RT but also no observable phosphorescence even at 77 K (**a, b, d**), and the excitation intensity drops rapidly below 370 nm (**c, d**).

Figure R14. a, b, Steady-state (Ss) PL spectra at 298 K and delayed ($\Delta t = 0.1$ ms) emission at 77 K of MNNI in PMMA (0.1% w/w). c, Excitation spectra (Ex) at 298 K. d, Steady-state (Ss) PL and excitation spectra at 298 K and delayed ($\Delta t = 0.1$ ms) emission at 77 K of RS-4FMNNI in PMMA (0.01% w/w).

Figure R15 shows spectra of MNNI doped in DBrB (4,4'-Dibromodiphenyl ether), a control serving as dibrominated solid-state host, still fails to produce RTP (**b, c, d**), even though DBrB contains two heavy Br atoms, indicating that the external heavy-atom effect is not the main cause of the RTP observed in the guest-host system.

Figure R15. a, Structures of guest molecules MNNI and a new host DBrB. Steady-state (Ss) PL and excitation spectra and delayed ($\Delta t = 0.1$ ms) emission at 298 K of RS-4FMNNI (b, $\lambda_{\text{ex}} = 420$ nm) and RS-PgMNNI (c, $\lambda_{\text{ex}} = 425$ nm) doped in DBrB (0.01% w/w). d, Enlarge of delayed emission.

Comment 2: The authors claimed that the red emission in Figure 1f originated from phosphorescence of the guest molecules, and one can see that the emission shows apparent vibronic structure. However, in comparison to the delay emission spectra shown in Figure S7(d), the emission pattern is drastically different.

Author Reply 2: We thank the reviewer for the pointing out this spectroscopic inconsistency. We zoom in these spectra lines from Figure S7d (i.e. delayed emission in CH_2Cl_2 , red line in **Figure R16a**) and compare them with delayed emission in 1,6-diiodohexane (DIH) and mTHF: we found that the delayed emission in CH_2Cl_2 is actually the emissive triplet state of 4FMNNI aggregates. The deeper theoretical analysis is detailed in a *Chem. Rev.* paper by the leading expert in molecular aggregates excitons, Prof. Frank

Spano from Temple University

(<https://pubs.acs.org/doi/abs/10.1021/acs.chemrev.7b00581>). The simpler version is that certain types of molecular aggregates completely inhibit the 0-0 band of the emissive state but enhances the 0-1 and/or 0-2 band intensity. The CH₂Cl₂ solution was recently shown by our group (<https://onlinelibrary.wiley.com/doi/10.1002/anie.202206366>) to induce molecular aggregates formation even at a very low concentration of 10⁻⁵ M at 77 K (but the benefit is that aggregates are good for enhancing phosphorescence according to Kasha's exciton model). As a result, we have provided the phosphorescence spectrum collected from heavy-atom substituted DIH (diiodohexane) that does not induce aggregates formation and the spectrum indeed shows both 0-0 and 0-1 bands, as in **Figure 1f**. Comparisons of guest emission in DIH with red emission in steady-state and delayed state are shown in **Figure R16c** and **e**, they have identical spectral profile and vibronic structures, confirm the red emission in **Figure 1f** exactly originates from the phosphorescence of the guest molecules. The same goes for Pg series shown in **Figure R16b, d, and f**. Although mTHF is the best solvent for obtaining discrete molecular phosphorescence spectrum at low temperatures, it fails to produce so due to the inherently low ISC efficiency of MNNI (**Figure R16**).

Figure R16. Delayed ($\Delta t = 0.1$ ms) emission and of RS-4FMNNI (a, $\lambda_{\text{ex}} = 420$ nm) and RS-PgMNNI (b, $\lambda_{\text{ex}} = 425$ nm) in mTHF, CH₂Cl₂ and DIH at 77 K. c, Comparison of a with Figure 1f (steady-state photoluminescence spectra of R-4FMNNI@R-4FBrBI) and Figure 1g (Delayed emission spectra of R-4FMNNI@R-4FBrBI). d, Comparison of b with Figure 4c and Figure 4d. e, f, Comparison of normalized spectra.

Comment 3: *Continuing comment #2, it is also critical to determine where the blue emission come from. Although the blue emission exhibits nanosecond population decay, which may support its assignment as fluorescence, upon exciting the host-guest mixtures at 365 nm, the guest molecules would be directly excited and display fluorescence as well. However, the emission onsets of them (Figure S7) are different from the blue band region in Figure 1f. Therefore, more likely, the blue band emission comes from the host since their onsets are the same (see Figure S7(c)). Also, according to my rough calculation, the energy transfer efficiency of the host is estimated to be ~100%. Accordingly, there should be nearly no population for the host in the excited state (triplet state) to observe the corresponding phosphorescence. Excitation spectra monitoring at the dual emission bands are absolutely necessary and the authors should check if the excitation and absorption spectra resemble in spectral profile and onset.*

Author Reply 3: We thank the reviewer for the good question. And the reviewer's analysis is correct. The blue band region in Figure 1f mainly originates from the **guest fluorescence**, but also contains the **host luminescence** (the host exhibits phosphorescence emission in DCM, mTHF, and DIH at 77 K, although the quantum yield in the solid state is quite low < 2%), as shown in **Figure R17a and R17b**. Consequently, when monitored with time-resolved decay, the blue band region of the doped sample exhibits both nanosecond (**c**) and millisecond decay kinetics (**d**), due to the **guest fluorescence** (Figure S22a) and **host phosphorescence** decay (Figure S23c). We agree with the reviewer that the triplet-triplet energy transfer should be very efficient for the doped sample. However, this does not necessarily mean the complete depletion of excited-state population of the host. This is because the triplet decay of the guest is very slow and thus the **triplet population of the guest gets saturated very quickly**. That said, there are no longer ground-state guest molecules left to accept more guest triplet exciton migration from the host, so that host phosphorescence can still be observed. In **Figure R17c**, the high baseline indicates that there could be triplet-triplet annihilation and other non-linear processes contributing to the guest fluorescence, given the oversaturated triplet excitons in the system. In fact, thermally activated delayed fluorescence (TADF) of guest molecules also contribute to the blue band region in Figure 1f, as shown in the temperature-dependent delayed emission (Figure S15). We have a follow-up study dedicated to TADF of doped systems being prepared. In

summary, steady-state emission consists of complex components from guest fluorescence, host luminescence from both monomers and aggregates in the solid state, non-linear luminescence and TADF from the guest, which render the overall spectral profile very complex. Incidentally, the luminescence studies of doped organic systems were very prevalent in the 1960s and 1970s, during which time many theoretical models were developed. However, contradicting reports and studies were also frequently published, highlighting the difficulty in analyzing these condensed phased organic molecules. On the other hand, we believe that the field could be re-examined with better purification and characterization techniques half a century later.

Figure R17. a, Comparison of Figure 1f with guest emission in DCM at RT, host emission in DCM at 77 K, and host solid emission at RT. b, Normalization of a. Time-resolved emission decay curves of the blue band region of R-4FMNNI@R-4FBrBI in Figure 1f, monitored with fluorescence mode (c, excited with nanoLED-370) or phosphorescence mode (d, spectraLED-370).

As per the reviewer's suggestion, steady-state excitation spectra monitoring at the blue band region (495 nm) and the red band region (615 nm) are shown in **Figure R18**. We notice that the blue band contains excitation from **both guest and host contributions**, consistent with our prior analysis. Again, the blue emission (blackline) and RTP (redline) come from completely different origins: while the blue band mainly stems from direct excitation of the guest, the red band (RTP) largely comes from the host. The results are also consistent with our Reply to Comment 1.

Figure R18. Normalized steady-state excitation spectra of R-4FMNNI@R-4FBrBI (10 ppm doped sample in Figure 1f) monitoring at the blue band (495 nm) and red band region (615 nm) compared with guest monomer excitation in DCM and host solid excitation spectra.

Comment 4: *In Figure S8(c) and S8(d), owing to the large Stokes shift and heavy-atom effect, the host molecules should exhibit phosphorescence in 77K. However, it is very weird that the prompt and delay ($t=0.1\text{ms}$) emission spectra display different emission onset.*

Author Reply 4: We thank the reviewer for the careful review and the important question. We have repeated the experiment and have replaced it with a new spectrum in Figure S8(c) and S8(d). We compiled the new prompt and the delay ($t=0.1\text{ms}$) emission spectra from Figure S8(c) and S8(d) in one Figure (**Figure R19**). We noticed that the onsets for them

are 419 nm and 423 nm, respectively. We attribute the visual discrepancy as the elevated baseline from severe scattering of the highly crystalline CH_2Cl_2 medium at 77 K. As can be seen that the pure solvent baseline is reduced to close to zero in delayed mode, suggesting that the discrepancy is not from impurity interference. This does not invalidate our conclusion.

Figure R19. Magnified view of steady-state (Ss) and delayed emission (DE) spectra of the host molecules in CH_2Cl_2 and pure CH_2Cl_2 solvents.

Comment 5: *The above concerns that I raised are all serious enough to affect the proposed mechanism. And the I strongly encourage the authors to carry out not only the emission spectra but also their corresponding excitation spectra before making any spectral assignment, so that the proposed mechanism can be clarified, to a large extent.*

Author Reply 5: We thank the reviewer for the extremely good suggestions, and we have gone to great length to address all the concerns. We hope that we have presented a very strong case to get the green light.

Reply to the report of the Reviewer 2

General comment: *In this manuscript, the authors report observation of unprecedented enantioselective RTP enhancement (CPE) values with 100 at room temperature from the organic chiral solid designed by them. This is the very good results I agree, which should be suitable for Nature Commun. The authors designed two chiral amino compounds naphthalimides R-4FMNNI and S-4FMNNI as guest molecules and two host compounds (R-4FBrBI and S-4FBrBI). For definition of Host and Guest molecules here might confused the audience, So the authors are suggested to give the clear distribution.*

Author Reply: We greatly appreciate the positive comments from the reviewer on our work. Our point-by-point response is given below. As for the possible confusion of Guest/Host definitions, we have added the following texts in the introduction part: “*It has to be noted that the phrase “host” in this study is referred to as an excited-state energy donor, the majority species, and “guest” the acceptor and minor species.*”

The Mechanistic Investigations is much interesting. The assembly of host-guest complex might be important to explain their observation of unprecedented enantioselective RTP enhancement. Therefore, this referee suggests the authors might discuss this assembly structure of host-guest complexes.

Author Reply: We thank the reviewer for the suggestion that the assembly structure should be discussed. However, this has been an unobtainable challenge ubiquitously faced in this field as of today: at a molar ratio of the order of 0.1% or less in the molecular solids or crystals, 1) X-ray crystallography techniques yielded identical results to those obtained for pure host solids (Figure S18); and 2) during the time of revision, we also attempted more advanced spectroscopic techniques such as sum frequency generation (SFG) spectroscopy and we did not extract any useful information with respect to the structure of the guest-host system. Of course, high-precision DFT calculations may be an alternative to address the problem, unfortunately, the simulation job at this scale with potentially tens of molecules at least is a more appropriate task for a long-term project after we are able to find a suitable theoretician collaborator. As an initial report on the story of its own kind, we are confident

that the current level of understanding is more than sufficient to excite the readers. And we are committed to getting to the bottom of the details in the near future.

Its application is very interesting too. the authors give the elementary study to recognize Chiral amino alcohols. this referee realized that an RTP enantioselective enhancement of derivatives of Chiral amino alcohols is about 10-folds with longer lifetime, it is a good result of course, but less exciting than the above discussed molecules. This referee is wondering if this RTP enantioselective enhancement of host-guest molecules system would be common rules? How about amino acid derivates?

Author Reply: We greatly appreciate the reviewer for the suggestion and we have thus synthesized the following amino acid sensing schemes (**Figure R20** and **R21**). As expected, the CPE method appears to be universal for distinguishing these chiral molecules. However, we believe that the results are redundant to this manuscript and decide to publish the data in a follow-up study with a central focus on amino acid sensing with different electronic donor and acceptor structures. We hope the editor/reviewer can understand our research plans.

Figure R20. a, structures of host L-PheBrBI and guests L-PheNI and D-PheNI. b, delayed ($\Delta t = 0.1$ ms) emission of L-PheNI and D-PheNI doped (0.01% doped) at L-PheBrBI (L@L, D@L) at 298 K ($\lambda_{\text{ex}} = 365$ nm). (*unpublished data*)

Figure R21. a, structures of hosts and guests. Steady-state (b, $\lambda_{\text{ex}} = 298 \text{ nm}$) and delayed emission (c, $\lambda_{\text{ex}} = 285 \text{ nm}$) of doped samples and host or guest at 298 K. d, Time-resolved emission decay curves of doped samples at 298 K (excited with spectraLED-280). (unpublished data)

Comment 1: In the end, In Fig 1, R@R, S @R et al are very confusing to the audience, which might be revised to the clear description R-molecular name or S=Molecular name as example.

Author Reply 1: We thank the reviewer for pointing out the possible confusion and we have tried changing the manuscript throughout to change R and S to corresponding abbreviations of chemical compound names. Unfortunately, the lengthy names render the figures less aesthetically appealing, which leads us to believe that the current nomenclature is perhaps better as is. An example is given below:

f**g****h**
Reviewers' Comments:

Reviewer #1:

Remarks to the Author:

The authors reply the reviewer's comments and suggestion, in part. However, there are still some problems regarding the proposed mechanism. These need to be addressed/solved. Otherwise, I still perceive rather weak support of this work from the fundamental basis. Listed below are comments raised relative to the authors' reply of my previous comments item by item.

Comments to the reply of previous comment #1:

- i. The authors should provide the solid-state absorption spectra in supporting information.
- ii. Indeed, provision of the excitation spectrum is one of the golden criteria for energy transfer process. The other criterion is to probe the lifetime of the acceptor and the donor, respectively. When adding the additional de-excitation pathway, the original population decay should be reduced. However, in figure S23, only rather slim difference in decays can be observed. Contradictorily, figure S23a and c even show the opposite trend, i.e., R@R displays longer lifetime when probing 486 nm.
- iii. Continuing ii., the overall energy transfer mechanism is proposed to be from triplet state of the host. The authors postulate that the intersystem crossing is the dominant de-excitation pathway of the S1 state owing to the heavy atom effect, and claim that this observation is consistent with barely emission at RT, i.e., the dominant non-radiative decay rate. However, logically, dominant non-radiative decay rate is not equivalent to dominant intersystem crossing. There are myriad of non-emissive materials in which the dominant non-radiative decay rate does not originate from intersystem crossing. Therefore, there is no solid evidence to prove the authors' assertion at all. In SI the authors claim that intersystem crossing efficiency in monomer is high in solution as well. Therefore, the direct evidence is to probe singlet oxygen emission or by chemical trapping that is very facile nowadays, and then derive the Φ_{ISC} . This will let the proposed mechanism more convincing. By the way, the authors mentioned that solid state is too complicated to simplify the mechanism. Then how about trace of trapping sites in solid? The following paper has proved that only trace amount of dopant can obtain long persistent luminescence.¹
 1. Nature 550, 384–387 (2017)
- iv. Explain why the energy transfer process in the proposed mechanism is a Dexter type? The distance between donor and acceptor should be provided. If the distance is greater than 10 angstroms, the Dexter type energy transfer involved in the mechanism is not quite possible.

Comments to the reply of previous comment #3:

The low doping concentration excludes the possibility of TTA. Furthermore, the energy difference between S1 and T1 is too large to exhibit TADF. Therefore, it is of great possibility that the floating baseline in figure R17c is simply due to re-excitation of the sample high excitation repetition rate.

Reviewer #2:

Remarks to the Author:

This is the revised version of previous review manuscript. The authors have carefully answered all of the questions raised by reviewers. It is deserving to say that the reported results of guest-host organic chiral solid (OCS) systems of chiral imide molecules, and observation of unprecedented enantioselective RTP enhancement (CPE) values from the OCSs are excited, although the this chirality-dependent energy-transfer (CDET) mechanism still need more investigated further. As the communication for rapid publication, this referee recommends the present manuscript to be published on the Journal Nature Commun.

Reviewer #1 (Remarks to the Author):

The authors reply the reviewer's comments and suggestion, in part. However, there are still some problems regarding the proposed mechanism. These need to be addressed/solved. Otherwise, I still perceive rather weak support of this work from the fundamental basis. Listed below are comments raised relative to the authors' reply of my previous comments item by item.

Comments to the reply of previous comment #1:

i. The authors should provide the solid-state absorption spectra in supporting information.

Author Reply: We thank the reviewer for this suggestion and we have added this Figure in the Supporting Information as well. This figure is now Figure S26 in the Supporting Information.

ii. Indeed, provision of the excitation spectrum is one of the golden criteria for energy transfer process.

Author Reply: In fact, donor (host) -dependent excitation spectrum of the acceptor (guest) RTP is **sufficient evidence** to prove energy transfer. If the reviewer is able to disprove or has an alternative explanation for this crucial piece of experimental evidence, it is definitely welcomed!

The other criterion is to probe the lifetime of the acceptor and the donor, respectively. When adding the additional de-excitation pathway, the original population decay should be reduced. However, in figure S23, only rather slim difference in decays can be observed. Contradictorily, figure S23a and c even show the opposite trend, i.e., R@R displays longer lifetime when probing 486 nm.

Author Reply: I would say that this is an iffy situation, and ideally, yes. What the reviewer suggested were mostly true if only the two emission bands could be completely separated. For instance, the energy transfer from a triplet state ligand to an f-f transition state of a rare earth element, a narrow atomic spectrum line of Eu ion for instance. For organic systems with broad-band emission, it is almost impossible to separate donor and acceptor emission, and hence, the unreliable change if probing lifetime instead.

iii. Continuing ii., the overall energy transfer mechanism is proposed to be from triplet state of the host. The authors postulate that the intersystem crossing is the dominant de-excitation pathway of the S1 state owing to the heavy atom effect, and claim that this observation is consistent with barely emission at RT, i.e., the dominant non-radiative decay rate. However, logically, dominant non-radiative decay rate is not equivalent to dominant intersystem crossing. There are myriad of non-emissive materials in which the dominant non-radiative decay rate does not originate from intersystem crossing. Therefore, there is

no solid evidence to prove the authors' assertion at all.

Author Reply: We understand that non-emissiveness is not equivalent to crossing over to the triplet manifolds.

In SI the authors claim that intersystem crossing efficiency in monomer is high in solution as well. Therefore, the direct evidence is to probe singlet oxygen emission or by chemical trapping that is very facile nowadays, and then derive the Φ_{ISC} . This will let the proposed mechanism more convincing.

Author Reply: We thank the reviewer for this suggestion. We have done so in DMSO solution using a singlet-oxygen capture agent 2,2,6,6-Tetramethylpiperidine (TEMP), and we have shown that under the 365-nm irradiation, the EPR signal of the capture agent "turned on" in the presence of the host molecule. The figure is now Figure S25 in the Supporting Information, which is mentioned in the main text highlighted as "(the presence of which is indicated in singlet-oxygen sensitization experiment shown in Figure S25)" on page 5.

Figure R1. EPR spectra of the host molecule **RS4FBrBI** in the presence of a singlet-oxygen capture agent 2,2,6,6-Tetramethylpiperidine (TEMP) before and after exposure to a broad-band 365-nm LED lamp.

By the way, the authors mentioned that solid state is too complicated to simplify the mechanism. Then how about trace of trapping sites in solid? The following paper has proved that only trace amount of dopant can obtain long persistent luminescence.¹

1. Nature 550, 384–387 (2017)

Author Reply: We thank the reviewer for point out the work. Trapping, in essence, is also a type of energy transfer process, which is not well-understood in both inorganic and organic systems alike.

iv. Explain why the energy transfer process in the proposed mechanism is a Dexter type? The distance between donor and acceptor should be provided. If the distance is greater than 10 angstroms, the Dexter type energy transfer involved in the mechanism is not quite possible.

Author Reply: We thank the reviewer for this question. The simple answer is that triplet-triplet energy transfer cannot proceed with the Förster type, given the vanishingly small transition dipole moments. And there is no way that the donor and acceptor molecules are separated by more than 10 angstroms (1 nm) in the crystalline state!

Comments to the reply of previous comment #3:

The low doping concentration excludes the possibility of TTA. Furthermore, the energy difference between S1 and T1 is too large to exhibit TADF. Therefore, it is of great possibility that the floating baseline in figure R17c is simply due to re-excitation of the sample high excitation repetition rate.

Author Reply: We understand the reviewer's concern. However, the temperature dependent delayed emission ($\Delta t = 1$ ms) spectrum does suggest TADF. It can be clearly seen from **Figure R2**, that, as the temperature increases, the intensity ratio between the fluorescence band and the phosphorescence band continuously increases as well.

Figure R2. Temperature-dependent delayed emission ($\Delta t = 0.1$ ms) spectra of **R-4FMNNI@R-4FBrBI** (10 ppm, $\lambda_{\text{ex}} = 424$ nm), from 80 K to 400 K with 10-K intervals.

Again, we would like to express that the solid-state organic molecules in the condensed phase is probably one of the most complex systems on the planet to physically model. There are numerous tiny phase changes and vibration-assisted reversible intersystem crossing, which are yet to be elucidated in the future.

Reviewer #2 (Remarks to the Author):

This is the revised version of previous review manuscript. The authors have carefully answered all of the questions raised by reviewers. It is deserving to say that the reported results of guest-host organic chiral solid (OCS) systems of chiral imide molecules, and observation of unprecedented enantioselective RTP enhancement (CPE) values from the OCSs are excited, although the this chirality-dependent energy-transfer (CDET) mechanism still need more investigated further. As the communication for rapid publication, this referee recommends the present manuscript to be published on the Journal Nature Commun.

Author Reply: We thank the reviewer for the understanding and we strive to get to the bottom of things persistently in the next few years!

Reviewers' Comments:

Reviewer #1:

Remarks to the Author:

Comments to the reply of previous comment #1-2:

Indeed, just as the authors' claimed that the organic systems with broad-band emission is complicated to probe the lifetime of individual components. However, my question mainly lies in the fact that whether the host transfers its excitation energy to the guest can be probed by the multiple component lifetime analyses. Note that the authors stated that the whole energy transfer mechanism stems from Dexter type throughout text. However, the 486 nm as the monitored wavelength for probing energy transfer in current measurement is a bad choice that provides no extractable information (Figure S23) because of strong mixing between donor and acceptor emission. According to energy-transfer mechanism, the lowest energy emission band should be ascribed to the phosphorescence from the final guest acceptor. The authors are suggested to monitor at the donor emission band to see the relaxation dynamics (quenching) due to the energy transfer. In this case, the authors are encouraged to monitor at the higher energy site of the donor emission to avoid the interference from the guest emission.

To summarize my points, it is necessary to display the correct trend of lifetime data and calculate all the energy transfer efficiency among all the reported mixture complexes. Let me emphasize again. To ensure the energy transfer process, from the steady-state approach, the excitation spectra must resemble the absorption spectra of the donor. From the dynamics point of view, the lifetime of the donor emission should be quenched due to energy transfer after mixing with the acceptor. Logically, it is impossible to meet only one of the above requirements, but not the other and still claimed that it is unambiguously ascribed to energy transfer process.

Comments to the reply of previous comment #1-3:

Also, the authors should carefully check whether the redox potential of the reported host molecules is higher than the trapping agent (TEMPO). If yes, the EPR signal of TEMPO would be present even if there is no singlet oxygen produced in the system. The attached paper provides the solid evidence toward this issue. The authors should provide either the cyclic voltammetry of the reported host molecules or singlet oxygen quantum yield.

Reference:

Nardi G, Manet I, Monti S, Miranda MA, Lhiaubet-Vallet V. Scope and limitations of the TEMPO/EPR method for singlet oxygen detection: the misleading role of electron transfer. *Free Radic Biol Med.* 2014 Dec; 77:64-70

Reviewer #2:

Remarks to the Author:

This is the 2nd revised version of previous review manuscript. I have read it again, and the authors have carefully answered all of the questions raised by reviewers. Considering of the proposed mechanism, I prefer to think it would be the Dexter energy transfer mechanism more probably. However, this referee also suggests the authors might add their explanation in the Main Text of revised manuscript. (triplet-triplet energy transfer cannot proceed with the Förster type, given the vanishingly small transition dipole moments. What is more, the donor and acceptor molecules are separated by less than 10 angstroms (1 nm) in the crystalline state, for example). By the way, when this referee checks the supporting information, the following suggestion is shown as below:

1. The procedures for Pure R-4FBrBI first doped with S-4FMNNI, and then doped with R-4FMNNI might be provided in the sample preparation in detail.
2. In the Synthetic procedure, the authors are suggested to check the reaction template description.
3. The purification of target compounds by silica gel chromatography, the (mixture) solvent as eluent are suggested to provide.
4. For NMR data. DMSO changed to DMSO-D6, or d-DMSO.

Reply to Reviewers' Comments for Manuscript NCOMMS-22-41250 for Nature Communications

Reviewer #1 (Remarks to the Author):

Comments to the reply of previous comment #1-2:

Indeed, just as the authors' claimed that the organic systems with broad-band emission is complicated to probe the lifetime of individual components. However, my question mainly lies in the fact that whether the host transfers its excitation energy to the guest can be probed by the multiple component lifetime analyses. Note that the authors stated that the whole energy transfer mechanism stems from Dexter type throughout text. However, the 486 nm as the monitored wavelength for probing energy transfer in current measurement is a bad choice that provides no extractable information (Figure S23) because of strong mixing between donor and acceptor emission. According to energy-transfer mechanism, the lowest energy emission band should be ascribed to the phosphorescence from the final guest acceptor. The authors are suggested to monitor at the donor emission band to see the relaxation dynamics (quenching) due to the energy transfer. In this case, the authors are encouraged to monitor at the higher energy site of the donor emission to avoid the interference from the guest emission.

To summarize my points, it is necessary to display the correct trend of lifetime data and calculate all the energy transfer efficiency among all the reported mixture complexes. Let me emphasize again. To ensure the energy transfer process, from the steady-state approach, the excitation spectra must resemble the absorption spectra of the donor. From the dynamics point of view, the lifetime of the donor emission should be quenched due to energy transfer after mixing with the acceptor. Logically, it is impossible to meet only one of the above requirements, but not the other and still claimed that it is unambiguously ascribed to energy transfer process.

Author Reply: We thank the reviewer's persistence and we have taken one step further to show *irrefutable evidence* for our proposed the "chirality-dependent energy transfer (CDET)" mechanism. This time, we monitored the (albeit weak) time-dependent phosphorescence emission at 420 nm of these samples (Figure R1), instead of at 486 nm

in the initial experiment. A brief summary of the results is that: 1) the pure chiral host (e.g., R) exhibits a single-exponential decay kinetics at 420 nm but shows substantial quenching kinetics in the presence of trace guest of the same chirality (e.g., R@R, 10 ppm); 2) a slight reduction in lifetime (plus bi-exponential decay) could be observed when the guest is of a different chirality (e.g., S@R, 10 ppm). The experimental results at both 77 K and room temperature gave similar trends. However, lifetime data analyses become much easier using results at 77 K, where a single-exponential decay could be obtained for the pure host sample. We believe that the results should support our claim “*that it is unambiguously ascribed to energy transfer process*” as pointed out by Reviewer 1.

Figure R1. Time-resolved emission decay curves of Pure Host (*R*-4FBrBI), *R*@*R* (10 ppm) and *S*@*R* (10 ppm) at 77 K (a) and at room temperature (b) when monitoring at 420 nm and using excitation with spectraLED-340 (340 nm).

Table R1. Photoluminescence properties of dopant samples at (10 ppm w/w) and pure host solids at 77 K and room temperature when monitored at 420 nm.

Temperature	77 K		Room Temperature		
Samples ^[a]	τ (ms) ^[b]		τ (ms) ^[c]		Weighted average lifetime (ms)
R-4FBrBI	5.407	100%	5.407	0.088 13.67% 0.427 47.16% 1.367 39.16%	0.749
S@R	0.548 14.46% 5.103 85.54%		4.444	0.081 11.65% 0.363 49.43% 1.073 38.92%	0.606
R@R	0.408 13.91% 3.463 86.09%		3.038	0.077 13.40% 0.304 41.44% 0.961 45.16%	0.570

[a] Guest-host molecular solids (w/w 10 ppm). [b] Phosphorescence lifetime at 77 K when monitored at 420 nm and excited with spectraLED-340. [c] Phosphorescence lifetime at room temperature when monitored at 420 nm and excited with spectraLED-340.

Meanwhile, as suggested by Reviewer 2, we have now included relevant discussions in the main text of the revised manuscript.

In Page 5, it now reads “To experimentally verify the proposed CDET mechanism, we measured the phosphorescence lifetimes of various samples at both 77 K and at room temperature (Figure S25), where the decay kinetics unambiguously reveal how host triplet excitons can be more selectively depleted by guest molecules of the same chirality but are less unaffected by guest molecules of the opposite chirality. As can be seen from Figure S25a, the pure host of **R-4FBrBI** exhibits single-exponential decay kinetics monitoring at 420 nm (phosphorescence emission belonging to **R-4FBrBI**) with an apparent lifetime of 5.4 ms at 77 K. However, the phosphorescence lifetime is substantially reduced when a trace amount (10 ppm) of guest **R-4FMNNI** is present, yielding bi-exponential decay kinetics (Table S6, $\tau_1 = 0.4$ ms and $\tau_2 = 3.5$ ms) with a pre-exponential-weighted average lifetime of 3.0 ms. To show that the energy-transfer processes is indeed chirality-dependent, we also measured the decay kinetics of **S-4FMNNI@R-4FBrBI** (10 ppm) monitoring at 420 nm. Surprisingly, the decay kinetics are more similar to those found for the pure host **R-4FBrBI** (Figure S25a), suggesting less effective triplet-triplet energy-transfer processes.

In addition, these results also give direct evidence for the Dexter-type CDET, since a dominant long-range Förster ET should not distinguish chirality, a short-range effect in this case. Furthermore, it is well known that triplet-triplet energy transfer cannot proceed with the Förster type, given the vanishingly small transition dipole moments.²⁷ In the current experimental condition (crystalline state), the donor and acceptor molecules are clearly separated by less than 1 nm, also creating the necessary condition for the Dexter process. We also performed the same measurements at room temperature. Although triplet-state quenching was also noted by guest molecules, the apparent decay kinetics for these samples are non-exponential (Figure S25b), which is not unusual for organic solids at elevated temperatures. We attribute the observed complexity at room temperature to enhanced molecular motions and various back-population (i.e., reverse intersystem crossing) pathways.”

Comments to the reply of previous comment #1-3:

Also, the authors should carefully check whether the redox potential of the reported host molecules is higher than the trapping agent (TEMP). If yes, the EPR signal of TEMPO would be present even if there is no singlet oxygen produced in the system. The attached paper provides the solid evidence toward this issue. The authors should provide either the cyclic voltammetry of the reported host molecules or singlet oxygen quantum yield.

Reference:

Nardi G, Manet I, Monti S, Miranda MA, Lhiaubet-Vallet V. Scope and limitations of the TEMPO/EPR method for singlet oxygen detection: the misleading role of electron transfer. *Free Radic Biol Med.* 2014 Dec; 77:64-70

Author Reply: We thank the reviewer for the thoughtful experimental details and we have now included the cyclic voltammetry (CV) of the host molecule and the trapping agent (TEMP) below. As shown in Figure R2a, the host molecule's redox potential is around -1.2 V. On the other hand, the CV of TEMP shows a monoelectronic, irreversible oxidation peak around 1.3 V, which is consistent with previous reports on secondary amines (e.g., Adenier et al. *Langmuir*, 2004, 20, 8243-8253.). Thus, the redox potential of the host molecule is lower than that of the trapping agent (TEMP).

Cyclic voltammetry was performed with an electrochemical workstation (CHI760E, Shanghai Chenhua Instrument Co., Ltd) using a three-electrode electrochemical cell, consisting of a glassy carbon disk working electrode (0.07 cm², BASi), an Ag/Ag⁺ quasi-reference electrode (BASi) with 0.01 M AgBF₄ (Sigma) in acetonitrile, and a platinum counter electrode (ALS).

Figure R2. Cyclic voltammograms (CV) of the host molecule **RS-4FBrBI** (a) and 2,2,6,6-Tetramethylpiperidine (b, TEMP) with concentration of 10 mM in 0.1 M TBAPF₆/DMF with a glassy carbon working electrode (0.07 cm², BASi); Negative scan from -0.9 V to -2 V for **RS-4FBrBI** and 1.8 to 0.8 for TEMP; Scan rate 50 mV/s; Sensitivity 1e-4 A/V. TBAPF₆: Tetrabutylammonium Hexafluorophosphate.

Reviewer #2 (Remarks to the Author):

This is the 2nd revised version of previous review manuscript. I have read it again, and the authors have carefully answered all of the questions raised by reviewers. Considering of the proposed mechanism, I prefer to think it would be the Dexter energy transfer mechanism more probably. However, this referee also suggests the authors might add their explanation in the Main Text of revised manuscript. (triplet-triplet energy transfer cannot proceed with the Förster type, given the vanishingly small transition dipole moments. What is more, the donor and acceptor molecules are separated by less than 10 angstroms (1 nm)

in the crystalline state, for example). By the way, when this referee checks the supporting information, the following suggestion is shown as below:

1. The procedures for Pure R-4FBrBI first doped with S-4FMNNI, and then doped with R-4FMNNI might be provided in the sample preparation in detail.

Author Reply: We thank the reviewer for the kind suggestion and we have added the detailed sample preparation in the supplementary information (SI).

In Page 3, it now reads “In particular, for the three-component doped samples such as ones used in Figure 2d-g, a fixed quantity of the pure host *R*-4FBrBI (100.00 mg) was placed in a 5-mL vial, to which the dichloromethane solution of *S*-4FMNNI (0.1 mL, 0.1 mg/mL) was added, followed by the addition of a dichloromethane solution of *R*-4FMNNI of various concentrations, and was finished by adding dichloromethane to adjust the solvent amount of all samples to be exactly the same.”

2. In the Synthetic procedure, the authors are suggested to check the reaction template description.

Author Reply: We appreciate the reviewer for the kind suggestion and we have carefully checked and improved the reaction template description.

3. The purification of target compounds by silica gel chromatography, the (mixture) solvent as eluent are suggested to provide.

Author Reply: We thank the reviewer for the suggestion and we have provided all the eluent solvent information in the SI.

4. For NMR data. DMSO changed to DMSO-D6, or d-DMSO.

Author Reply: We thank the reviewer and have changed “DMSO” to “d-DMSO”.

Reviewers' Comments:

Reviewer #1:

Remarks to the Author:

The authors have replied and made corresponding changes in a satisfactory manner. I only have a small comment left regarding TableR1. In 77K and 298K, where do the short lifetimes of S@R and R@R stem from? The authors should specify the IRF of the given lifetime measurements. If the shorter lifetimes are beyond temporal resolution, then a tail-fit would be more appropriate.

Reply to Reviewers' Comments for Manuscript NCOMMS-22-33525B for Nature Communications

REVIEWERS' COMMENTS

Reviewer #1 (Remarks to the Author):

The authors have replied and made corresponding changes in a satisfactory manner. I only have a small comment left regarding Table R1. In 77K and 298K, where do the short lifetimes of S@R and R@R stem from? The authors should specify the IRF of the given lifetime measurements. If the shorter lifetimes are beyond temporal resolution, then a tail-fit would be more appropriate.

Author Reply: We thank the reviewer for the careful review. We believe that the short lifetimes are most likely due to heterodimer emission, which is known to accelerate excited-state decay processes based on previous studies (e.g., <https://www.nature.com/articles/s41467-019-13048-x>). The lifetime measurements were recorded on a single photon counting controller: Fluorohub (Horiba Scientific), and phosphorescence lifetime data were acquired with a LED laser (SpectraLED-340 in Figure R1) with the excitation peak at 344 nm and wavelength FWHM 9 nm. Lifetime data were analyzed with Data Station v6.6 (Horiba Scientific). In this instrument and data analysis station, phosphorescence decay mode requires NO prompt (IRF curve). We also tried and the instrument also is not allowed to measure IRF in phosphorescent mode (the instrument has No response when measure IRF in phosphorescent mode). According to the product introduction, *the SpectraLED is a novel light source designed specifically for the measurement of phosphorescence lifetimes, has No afterglow, permitting easier interpretation of lifetimes shorter than 100 μ s, thus the shorter lifetimes ($> 77 \mu$ s) are not beyond temporal resolution.* More details about SpectraLED in the official website link: <https://www.horiba.com/fra/scientific/products/detail/action/show/Product/spectraled-1095/>

Figure R1. Photo of SpectraLED-340: The LED laser used for phosphorescence lifetime measurements in Table R1 (with the excitation peak at 344 nm and wavelength FWHM 9 nm).

Incidentally, fluorescence lifetime data were acquired with a 1 MHz LED laser with the excitation peak at 372 nm (NanoLED-370), the prompt (IRF) curve of fluorescence decay mode, which is used in the fitting of fluorescence lifetime, was provided in all fluorescence lifetime spectra.